# Longitudinal Impact of WTC Dust Inhalation on Rat Cardiac Tissue Transcriptomic Profiles

**DOI:** 10.3390/ijerph19020919

**Published:** 2022-01-14

**Authors:** Sung-Hyun Park, Yuting Lu, Yongzhao Shao, Colette Prophete, Lori Horton, Maureen Sisco, Hyun-Wook Lee, Thomas Kluz, Hong Sun, Max Costa, Judith Zelikoff, Lung-Chi Chen, Matthew W. Gorr, Loren E. Wold, Mitchell D. Cohen

**Affiliations:** 1Department of Environmental Medicine, New York University Grossman School of Medicine, New York, NY 10010, USA; colette.prophete@einstein.yu.edu (C.P.); lajh1@hotmail.com (L.H.); gbsisco@optonline.net (M.S.); Hyun-wook.Lee@nyulangone.org (H.-W.L.); Thomas.Kluz@nyulangone.org (T.K.); Hong.Sun@nyulangone.org (H.S.); Max.Costa@nyulangone.org (M.C.); Judith.Zelikoff@nyulangone.org (J.Z.); lcc4@nyu.edu (L.-C.C.); Mitchell.Cohen@nyulangone.org (M.D.C.); 2Departments of Population Health & Environmental Medicine, New York University Grossman School of Medicine, New York, NY 10016, USA; Yuting.Lu@nyulangone.org (Y.L.); Yongzhao.Shao@nyulangone.org (Y.S.); 3Department of Physiology and Cell Biology, Dorothy M. Davis Heart and Lung Research Institute, The Ohio State University College of Medicine, Columbus, OH 13210, USA; gorr.1@osu.edu (M.W.G.); Loren.Wold@osumc.edu (L.E.W.); 4College of Nursing, The Ohio State University, Columbus, OH 13210, USA

**Keywords:** WTC dust, rat cardiac tissue, transcriptomic profiles

## Abstract

First responders (FR) exposed to the World Trade Center (WTC) Ground Zero air over the first week after the 9/11 disaster have an increased heart disease incidence compared to unexposed FR and the general population. To test if WTC dusts were causative agents, rats were exposed to WTC dusts (under isoflurane [ISO] anesthesia) 2 h/day on 2 consecutive days; controls received air/ISO or air only. Hearts were collected 1, 30, 240, and 360 d post-exposure, left ventricle total RNA was extracted, and transcription profiles were obtained. The data showed that differentially expressed genes (DEG) for WTC vs. ISO rats did not reach any significance with a false discovery rate (FDR) < 0.05 at days 1, 30, and 240, indicating that the dusts did not impart effects beyond any from ISO. However, at day 360, 14 DEG with a low FDR were identified, reflecting potential long-term effects from WTC dust alone, and the majority of these DEG have been implicated as having an impact on heart functions. Furthermore, the functional gene set enrichment analysis (GSEA) data at day 360 showed that WTC dust could potentially impact the myocardial energy metabolism via PPAR signaling and heart valve development. This is the first study showing that WTC dust could significantly affect some genes that are associated with the heart/CV system, in the long term. Even > 20 years after the 9/11 disaster, this has potentially important implications for those FR exposed repeatedly at Ground Zero over the first week after the buildings collapsed.

## 1. Introduction

In the years since the 9/11 disaster, chronic health problems continue to become evident among firefighters/rescue personnel (First responders; FR) who were at the World Trade Center (WTC) Ground Zero for repeated/prolonged periods during the first 72 h post-collapse [1,2]. Evidence from clinical studies and data accumulated by the WTC Health Registry have noted increases in the incidence of atherosclerosis and other heart diseases [3,4]. Even so, underlying reasons for the development of these disorders in the FR remain undefined. Moreso, to date, one major problem that has remained unresolved in these subsequent 20 years has been that underlying reasons for development of these disorders in the FR remain undefined. More critically, it has still not been established if WTC dust itself was a causative agent able to potentially alter the structure/function of the heart and associated vasculature in the exposed FR. Showing causality is important in that a majority of the exposed FR were not wearing proper personal protective equipment (PPE) or removed it due to the extraordinary conditions at Ground Zero when levels of the WTC dusts were at their highest. It still has not been established if WTC dust is a causative agent able to alter the structure/function of the heart and associated vasculature.

Determining if those WTC dusts predominant in the Ground Zero air in the first week post-9/11 were cardiotoxic of their own accord is important, as other pollutants that could mask or amplify any true dust effects increasingly became omnipresent in the air after that first week. The latter included known cardiotoxic diesel exhaust particles (DEP, from trucks and cranes involved in increased rescue/clean-up efforts). In general, Ground Zero air during the initial 72 h post-collapse period predominantly contained building-derived materials (collectively termed ‘WTC dust’). This WTC dust was a mixture of fine, coarse, and (mainly) alkaline super coarse particles (>95% of mass 10–53 μm [mass median aerodynamic diameter (MMAD) = 22 μm])—each bearing a wide variety of toxic metals and organics. Specifics on the composition of this first week dust have been extensively detailed in the previous publications [1,2,5,6,7]. WTC dust particle concentrations in the first 72 h were high; estimates for 9/11/01–9/13/01 were in the 100s of mg particles/m^3^, that ‘dropped’ to mg—hundreds of μg/m^3^ after ≈ 3 wk. As a result, it was deemed that the arrival time and respirator use/non-use during the initial 72 h, and persistence of exposures over the course of the first week post-disaster, were critical factors in the development of health problems in the FR [1].

Under normal circumstances, inhaled complex particles have a relatively limited period to induce toxicity. However, until cleared, many particle constituents can induce local/systemic toxicities [8]. Because inhaled alkaline agents can cause damage that leads to respiratory epithelium cell death, airway denudation, and altered cilia beating/ciliostasis [9,10,11], it is possible that entrained alkaline super coarse WTC particles could have caused altered clearance activities in the lungs of FR. Our previous studies showed that rats exposed to WTC dust (using exposures/levels mimicking FR mouth-breathing) led to persistent reductions in airway ciliated cell numbers and clearance of WTC dust itself [12]. Thus, apart from the WTC dust possibly being direct inducers of cardiovascular (CV) damage, these inhaled alkaline dusts could have led to reduced clearance from the lungs of other known CV toxicants that were inhaled by dust-exposed FR at later timepoints (such as DEP). If the dusts themselves are not causative of heart pathologies, this latter mechanism provides one additional novel means to potentially explain the increased incidence of CV problems seen in FR.

To begin to understand if WTC dusts could have potentially directly impacted on the hearts of exposed FR, evaluations of dust-induced longitudinal changes in gene expression in heart tissues would be useful. Clearly, it is not plausible to directly evaluate heart tissues from these FR. Thus, to ascertain if WTC dusts might have led to altered gene expression in the heart, this first-of-its-kind study used spontaneously hypertensive (SHR) rats as models for the FR. Our previous studies showed that the SHR is a valid model to assess health effects from WTC dust exposures [13]. In the present studies, SHR rats were exposed on consecutive days for 2 h/day (via intratracheal inhalation) to WTC dust collected on 12–13 September 2001, and transcriptome profiles were generated for their hearts collected at multiple timepoints over a 1 year post-exposures period.

No previous studies have undertaken an examination to see if the WTC dusts were causative agents for the documented increases in the incidence of atherosclerosis and other heart diseases in FR. While proof of causation will be obtained if any effects on gene expression in the rat hearts are found, such changes will ultimately need to be evaluated to see if they correspond to changes in cardiac/cardiovascular structures/functions. Ongoing studies unique to our laboratories (and those of collaborating investigators) are examining these latter parameters. In the end, how strongly any structural/functional changes in the heart can then be correlated to any changes in gene expression seen here will be presented in follow-on publications. Lastly, though not a major objective of this study, data obtained here may also allow for the potential identification of novel sero-markers of dust exposure-induced CV changes. Such markers could prove critical even now in that they could be used by clinicians to identify sooner any presence of CV defects in otherwise still-healthy, dust-exposed FR.

In summary, the objective of the present study was to—for the first time—demonstrate that realistic exposures to Ground Zero dusts could induce gene expression changes in the heart. Whether such changes correspond to actual changes in heart structure/functions are the subject of parallel ongoing studies in the same rat model. Of key interest to us was to see if any induced effects were long term in developing, as this would be most-relevant to exposed FR at this point 20 years after the disaster. Unlike the critical epidemiologic studies that have linked first week repeated Ground Zero dust exposures to increased cardiac/cardiovascular problems in FR, the studies here will stand alone as the first to potentially directly demonstrate causality of effects from the WTC dusts alone on the heart.

## 2. Materials and Methods

### 2.1. Animals

The SHR (spontaneously hypertensive) rat model was used to reflect any potential impact from exposures that FR faced when exposed to “unique” WTC dust at Ground Zero during the period of 11–13 September 2001. Selection of the SHR rat was based on several factors, i.e., it is used: (1) to model chronic hypertension that progresses to heart failure; (2) to study mechanisms of hypertension-induced hypertrophy that progresses to heart failure; (3) in studies pertaining to damage to arteries in general and athero-/arteriosclerosis; and, (4) in studies to assess inducible changes in CV structure/function/gene expression [14,15].

In the studies here, male SHR (10-week-old; Harlan Labs, Frederick, MD) were placed in polycarbonate cages with corncob bedding in a facility maintained at 23 °C with a 30–50% relative humidity and 12-h light/dark cycle. Food (Purina lab chow) and filtered tap water were provided ad libitum. All rats were acclimated 1 week prior to use. All animal procedures here were conducted under an animal protocol (Protocol #IA16-01467) approved by New York University SOM Institutional Animal Care and Use Committee (IACUC).

### 2.2. WTC Dust and Generation for Exposures

Dust was collected at representative sites on/around Ground Zero on 12–13 September 2001 [2,16,17,18]. All samples were stored in airtight containers in the dark at room temperature to minimize potential light, heat, or ambient gas-induced changes in dust physicochemical properties. Parent WTC dust was sieved to yield all particles of diameters ≤ 53 μm (i.e., WTC_53_, referred to hereafter as WTC dust) for use in exposures. Details about the preparation of the WTC_53_- materials, as well as physicochemical properties of this fraction, and the intratracheal inhalation integrated (ITIH) system used to expose the rats have all been previously reported [2,16,17,18].

Based on filter measurements taken before and immediately after each 2 h exposure (i.e., no concurrent measures due to impact on flow delivery to rats), the rats in the day 1, 30, 240, and 360 groups were exposed to the WTC dust at an average level of 32.22 [±1.48 (SE)] (range 29.96–35.45) mg dust/m^3^. This average dust level across all exposures was calculated to approximate one of ≈250 mg dust/m^3^ that was likely to have been experienced by a mouth-breathing “reference” FR and others at Ground Zero over a 4 h period during the initial 72 h after the buildings collapsed (Mayor’s WTC Medical Working Group; personal communication). The mass median aerodynamic diameter (MMAD) of the WTC dust study was confirmed (via horizontal elutriation prior to animal exposures) to be 23 μm (σ_g_ = 1.45). As the dusts used in each exposure were comprised of all particles of diameters that was ≤53 μm, the atmospheres introduced into the rat lungs were nominally comprised of more than >95% particles of 10–53 μm, 1.1% in the 2.5–10 μm range, and ≈3.5% as <2.5 μm.

### 2.3. Experimental Design

For technical reasons (i.e., to allow WTC particles, including those of d_a_ > 2.5 μm, to circumvent the rat nasal region so they were introduced and deposited in lungs in a manner to mimic mouth-breathing FR), it was necessary to use an ITIH system to expose the rats.

For the study, a group of randomly-selected rats (*n* = 5–6/group) were exposed for 2 h periods on two consecutive days to WTC_53_- dusts while under isoflurane anesthesia (ISO; IsoFlo, Abbott Laboratories, Chicago, IL, USA) in O_2_ carrier gas (2.5% final concentration after mixing with house air). As control groups, a second group of rats were exposed to ISO only (2.5% in carrier O_2_ gas) and a third group of rats were exposed to air only (Naïve). 

At various timepoints (Days 1, 30, 240, and 360) post-final exposure, rats in each of the three treatment groups were euthanized by injection with Sleepaway (500 mg/kg; Fort Dodge Animal Health, Fort Dodge, IA, USA). Hearts were immediately removed and weighed, and portions of the left ventricle (LV), left atrium (LA), right ventricle (RV), and right atrium (RA) were collected, flash frozen in liquid N_2_, and placed at −80 °C for later use.

### 2.4. Data Analysis: Overview

The main hypothesis here was that WTC dust alone may have significant effects on the hearts of exposed rats. As it was necessary to use ISO anesthesia when rats were exposed to the dusts, the WTC rats were actually co-exposed to dust and ISO concurrently. In an absence of any ISO effects, the ISO and Naïve rat expression patterns would be similar and any effect of WTC dust could then be detected by simply comparing WTC vs. Naïve rats or WTC vs. ISO rats. However, due to the existence of non-negligible short- and long-term effects from ISO vs. Naïve group of rats, Park et al. [19], it claims that effects attributable to the WTC dust could not be directly observed by using such simple comparisons. Short-term and long-term effects of ISO had to be accounted for to avoid falsely attributing ISO effects to WTC dust. Accordingly, the study design used both a naïve control and an extra ISO control. To discern effects attributable to WTC dusts, cross-sectional analyses of differential gene expression could then be conducted to compare WTC vs. Naïve, WTC vs. ISO, and ISO vs. Naïve rats at each timepoint (Days 1, 30, 240, and 360) post-second exposure.

As this study was interested in identifying possible effects of WTC dust in the rats, it was important to identify genes differentially expressed between the WTC dust and Naïve rats. Further, these analyses wanted to exclude the possibility that the observed differential expressions between WTC and Naïve rats were actually due to any effects from ISO. Therefore, any genes of interest were required to also be differentially expressed between the WTC and ISO rats. By this, effects attributable to WTC dust on rats could be determined when there was evidence of consistent effects for both WTC vs. ISO and WTC vs. Naïve rats, but in the absence of significant effects when comparing ISO vs. Naïve rats.

### 2.5. RNA-seq Analysis

RNA-seq protocols routinely performed at the NYU NIEHS center were used to analyze changes in the transcriptome (mainly including mRNA and long non-coding RNA) to identify key components targeted and the gene networks and pathways that may contribute to the adverse health effects/pulmonary pathologies associated with WTC dust-induced pathways [20,21]. In brief, total RNA from each rat LV was isolated using TRizol (Invitrogen, Waltham, MA, USA). Any DNA contamination was removed using a DNA-free kit (Invitrogen) according to the manufacturer’s protocols. From the final materials (whose purity and concentration were confirmed using formaldehyde gels and a Nano UV-Vis Spectrophotometer), RNA-seq libraries were prepared using a TruSeq RNA Sample Prep kit v2 (Illumina). Next, an Illumina HiSeq-4000 system (NYU Genome Technology Center) was used to perform sequencing of 1 × 50 single-end reads. Raw sequence data (Fastq) were loaded into CLC Genomics Workbench (Version 20.0.4 Qiagen) for data analysis. The raw Fastq files were trimmed to remove any remaining adaptors and ambiguous nucleotides. The trimmed sequence files were aligned to human genome (Hg38) allowing two mismatches. Reads mapped to the exons of a gene were summed at the gene level. Gene expression levels were quantified as total counts per million (TPM).

### 2.6. Differential Gene Expression Analysis

Genes with low expression values (counts per million lower than 0.5 across all groups) were excluded from the subsequent analysis. The RNA-seq read counts data were normalized by the trimmed mean of M-values (TMM) method after filtering out low expression values. Cross-sectional and longitudinal analyses of differentially expressed genes were conducted to compare the expression level between the WTC group and ISO group, WTC group and Naïve group, and ISO vs. Naïve group at each post-exposure timepoint (Days 1, 30, 240, and 360) using the R/Bioconductor software package edgeR in the R statistical programming environment [22,23]. 

### 2.7. Statistical Analysis

For each gene, the expression level was modeled by the generalized linear model and quasi-likelihood (QL) F-test was used to compare the gene expression level between any two groups. The logarithm of fold-change (logFC; to base 2) and nominal p-values of the F-test were obtained. The Benjamini–Hochberg false discovery rate (FDR) was used as an adjustment for multiple testing to avoid abundant false positive DEG that could arise due to testing a vast number of genes. Principal component analysis (PCA) was performed on normalized log-transformed expression values (after filtering out low expression values) for all samples to detect any possible outliers at each timepoint as shown in Appendix A (No evidence of outliers among the samples was found). The analyses here have also tried to identify potential batch effects across different timepoints by plotting gene expressions measured at different timepoints from the same tissue (Note: no evidence of batch effects was found).

To identify possible effects attributable to WTC dust, according to a commonly accepted standard on FDR, cross-sectional analyses of differentially expressed genes (DEGs) with FDR < 0.05 and same sign of logFCs both in comparing the WTC vs. ISO groups and in comparing the WTC vs. Naïve groups were conducted in the absence of significant effects when comparing ISO vs. Naïve rats. Given the relatively small sample size at each timepoint and the severe burden of the multiple testing adjustment, the current study also tried to find evidence of genetic effects attributable to WTC dust beyond those from ISO in the following two scenarios: (1) There are DEGs with FDR < 0.05 when comparing WTC vs. ISO rats, and these DEGs also have FDR ≤ 0.15 when comparing WTC vs. Naïve rats with logFC having the same signs in both comparisons; (2) There are DEGs with FDR < 0.10 when comparing WTC vs. ISO rats, and these DEGs also have FDR < 0.05 when comparing WTC vs. Naïve rats with logFC having the same signs in both comparisons. The DEGs satisfying either of the two criteria were also examined to confirm that their gene expressions in the ISO group are not significantly different from those of the naïve rats. That is, any DEGs whose expression level was greater/lower in the WTC group than in both the ISO and Naïve groups under a certain significance level in FDR, could potentially indicate effects attributable to the WTC dust alone in the absence of any significant ISO effects noted when compared against the Naïve rats.

To see how the WTC dust affected these DEGs throughout the time course, longitudinal trajectory was plotted to examine whether there are temporal trends for each of the selected DEGs, based on the logFC at all post-exposure timepoints.

In addition, this study investigated genes with differential expression between the WTC and ISO rats at day 360, but not significantly different between WTC vs. Naïve rats. For some of these genes, the ISO-only rats might have long-term protective effects but such effects might be cancelled by the WTC dust and not detectable in the comparison of Naïve vs. WTC rats (the latter which were in fact concurrently exposed to both dust and ISO) due to potential interactions between the ISO and WTC dust. That is, for these genes, one might see the differences between WTC and Naïve rats were not significant while they were between WTC and ISO rats. Thus, as one type of further exploratory analysis, this study investigated potential long-term effects of WTC dust via WTC–ISO interactions by studying genes with a p-value < 0.05 at day 360 and a p-value < 0.1 at day 240 for WTC vs. ISO comparisons and having the same sign of logFC at both days 240 and 360. To see if the selected set of genes were enriched in certain functional pathways, functional gene set enrichment analysis (GSEA) was conducted using metascape (http://www.metascape.org, accessed on 5 November 2021) [23].

## 3. Results

### 3.1. Cross-Sectional Analyses of Differentially Expressed Genes (DEGs) at Days 1, 30, 240 and 360

Only two genes, *Rps18* and *Surf1*, satisfied the condition of an FDR < 0.05 and a same sign of logFCs both in comparing the WTC vs. ISO and in comparing the WTC vs. Naïve groups of rats on day 360, and none at days 1, 30, or 240. In fact, no genes had an FDR < 0.15 at days 1 and 30 when comparing WTC vs. ISO rats, indicating that the WTC dust did not have a significant short-term effect beyond any induced by ISO. 

Even under the relaxed criteria (1) and (2), outlined above (see 2.7 Statistical analysis section), no DEGs were found under the two scenarios at days 1, 30, and 240. This indicates that WTC dust did not have a significant short-term effect beyond that of ISO. At day 360, 14 DEGs were identified under the two scenarios, where Rps18 and Surf1 had an FDR < 0.05 for both comparisons, 7 of the 14 DEGs with an FDR < 0.05 for WTC vs. ISO rats and an FDR ≤ 0.15 for WTC vs. Naïve rats, and 5 of them with an FDR < 0.10 for WTC vs. ISO rats and an FDR < 0.05 for WTC vs. Naïve rats. Table 1 lists the logFC, *p*-value, and FDR of the 14 DEGs. In addition, for these DEGs, there was no significant difference in expression levels when comparing the ISO vs. Naïve rats. Therefore, these 14 DEGs could potentially reflect effects at day 360 attributable to the WTC dust beyond any due to ISO. The majority of these DEGs have been implicated as having an impact on heart functions (see Discussion).

### 3.2. Longitudinal Analyses of the 14 Selected DEGs

The cross-sectional analysis of differential gene expression identified 14 DEGs with effects attributable to WTC dust at day 360. It is also of interest to examine how these genes would change throughout the entire time course. As a study designed with longitudinal measures of gene expression at multiple timepoints, the trajectories of logFC between WTC and ISO for the 14 DEGs were plotted in Figure 1, the trajectories of logFC between WTC and Naïve were plotted in Figure 2, where blue lines are trajectories of DEGs up-regulated at day 360, also labeled as ‘Up’, and red lines referred to the trajectories of DEGs down-regulated at day 360, labeled as ‘Down’. The bold lines connecting the triangles are the mean trajectories. The information presented in Figure 1 and Figure 2 suggest that the logFCs of the blue lines were not different from 0 at days 1 and 30, started to increase since day 240 and reached a peak at day 360. The red lines were not different from 0 at days 1 and 30, started to decrease since day 240, and reached a peak at day 360. In other words, these WTC effects appeared to be late onset and lasting long term. Figure 1 and Figure 2 each also show similar trends across time, implying the observed effects of the 14 DEGs are attributable to WTC dust rather than ISO. In addition, PCA analyses were conducted on the 14 DEGs using all samples at each day. The result in Appendix A showed that the 6 up-regulated and 8 down-regulated genes were not separable until day 360, which was consistent with the findings in the differential gene expression analysis and trajectory analysis.

### 3.3. Potential WTC Effects in the Presence of ISO-WTC Dust Interaction

No DEGs with an FDR < 0.05 between WTC vs. ISO rats were identified at days 1, 30 and 240, indicating that the WTC dust may not have significant short-term effects on rats (after multiple testing correction). Thus, the study focused on identifying long-term effects of the dust. At day 360, 14 genes were identified that have significant differential expressions between WTC and ISO and between WTC and Naïve rats (listed in Table 1 and Figure 1 and Figure 2). In addition to these 14 genes with long-term effects attributable to WTC dust alone, it is also possible that some genes might be affected in the long term by WTC dust via interaction effects during the combined WTC dust–ISO exposure. For example, it was seen that some genes though not significantly different between the WTC and Naïve groups, showed significant differences in expression between the Naïve and ISO as well as the WTC vs. ISO groups. Such genes are potentially of interest since they might suggest potential effects of WTC dust alone in the presence of interaction effects between ISO and WTC dust. For example, some genes in the ISO alone group might have some long-term protective effects and that those protective effects might be cancelled by the WTC dust in the WTC group due to any ISO–WTC dust interactions.

One might imagine that the WTC dust alone might have long-term effects based on evidence of these interactions though there could not be a ‘true’ group of rats with WTC dust alone due to technical challenges. Thus, this study further investigated potential long-term effects of WTC dust from the set of genes that have small p-values for WTC vs. ISO but not necessarily for WTC vs. Naïve, which might reflect ISO–WTC dust interactions. Given that most of these genes are significant when comparing ISO vs. Naïve rats, but do not have very small p-values when comparing WTC vs. ISO rats even at day 30, the current analysis was restricted to those genes with consistent expression pattern between days 240 and 360 so as to reduce the chance of too many false positive p-values. Therefore, from the analyses of differential gene expression comparing WTC against ISO rats on days 360 and 240, a set of 124 DEGs whose logFCs have the same sign on days 360 and 240 with *p*-value < 0.05 at day 360 and *p*-value < 0.1 at day 240 were identified. Among the 124 DEGs selected, 60 were up-regulated at both days 240 and 360, and 64 were down-regulated at both days 240 and 360. Results of the functional gene set enrichment analysis (GSEA) for the 60 up-regulated genes is shown in Figure 3; the data indicate that PPAR signaling pathway is the most significant pathway enriched by the 60 genes (and characterized by DEGs Aqp7, Fabp3, Rxrg, Ehhadh, Plin5). Results of enrichment analyses for the 64 down-regulated genes are shown in Figure 4; here, the data indicated that heart valve morphogenesis was the most significant pathway enriched by the 64 genes (and characterized by DEGs Bmpr1a, Tgfb2, Bmpr2, Emilin1, Dchs1). These top pathways and genes may also relate to heart functions as discussed below. The lists of DEGs with logFC and p-value are summarized in the Appendix A.

## 4. Discussion

Since the World Trade Center (WTC) towers collapsed on 11 September 2001, there has been growing concern over health effects among first responders (FR) who were on site 12–13 September 2001 and for many days thereafter. As the building materials were pulverized, the collapse resulted in the release of dense dust clouds of particles throughout Ground Zero as well as to areas around the original complex (and beyond). Epidemiologic studies of these FR have shown that those who were exposed to the dusts (along with other on-site pollutants such as diesel exhaust and metal cutting fumes) on 11 September, and over the first days thereafter, have increasingly presented with respiratory/cardiac pathologies. While adverse respiratory consequences in FR were seen within months of the disaster, over time, a large variety of respiratory and cardio-vascular pathologies have developed.

This is the first rodent study to analyze differential gene expression in the hearts as a result of WTC dust exposures. Here, spontaneously hypertensive (SHR) rats (now-validated models for the FR) were exposed to Ground Zero dusts (that were collected 12–13 September 2001) under paradigms that mimicked exposures/doses faced by on-site mouth-breathing FR who remained at/returned to Ground Zero over 11–13 September 2001 [16,18]. These investigations are also the only ones to use Ground Zero dusts collected within the first 72 h post-disaster and, thus, are “unadulterated”. Note: “Unadulterated” is used to reflect the composition of these dusts vs. all other WTC dust collected after a significant rainstorm on 9/14/01 and thereafter as the Main Pile fire raged [until December 2001]. No samples exist of dusts generated on 9/11/01; the samples used here represent the only existing materials that were on-site at Ground Zero and to which FR were repeatedly exposed in the critical initial 72 h period.

Here, to allow for the continuous dust exposures, rats were anesthetized with isoflurane (ISO), a commonly used inhalation anesthetic for laboratory experiments [24] and widely used on humans in some elective surgeries [25,26]. Though ISO is believed to maintain better cardiac function than other anesthetics (such as a combination of ketamine and xylazine), it can induce various adverse [27,28,29,30,31,32] and beneficial side effects [33,34,35,36,37,38,39,40,41]. Therefore, as the SHR rat was used to model potential WTC dust-induced effects in hearts of exposed FR, and potential cardiovascular effects from repeated exposures to ISO were unavoidable in the exposures here, any unintentional ISO-induced effects had to be carefully accounted for in all data obtained.

The study design used two control groups so that “pure” WTC dust effects relative to those from ISO alone could be discerned. Longitudinal measures of gene expression also allowed for any time-related effects from WTC dust (relative to ISO and to Naïve rats) to be evaluated as well. While genes that are differentially expressed between the WTC and Naïve groups of rats are of interest, those genes are also required to be differentially expressed between the WTC and ISO rats to avoid the possibility that the identified differential gene expression between WTC and Naïve rates are actually due to the effects of ISO alone. Consequently, attributable effects of WTC dust on rats can be determined when there is evidence of consistent effects for both WTC vs. ISO and WTC vs. Naïve rats and in absence of significant difference between ISO and Naïve rats. That is, the existence of DEGs whose expression level were significantly greater/lower in the WTC group than both the ISO and Naïve groups under a certain significance level in FDR.

To this end, differential gene expression analyses were conducted to compare WTC vs. Naïve, WTC vs. ISO, and ISO vs. Naïve rats. After using FDR for multiple testing correction, two genes (Rps18 and Surf1) satisfied an FDR < 0.05 and same sign of logFCs for both comparisons at day 360, and none at days 1, 30, or 240. Given the relatively small sample size and severity of the multiple testing adjustment, two scenarios were also considered to find evidence of attributable effects of the WTC dust: (1) there are DEGs with an FDR < 0.05 when comparing WTC vs. ISO rats, and these DEGs also have an FDR ≤ 0.15 when comparing WTC vs. Naïve rats—with each logFC having the same signs in both comparisons; (2) there are DEGs with an FDR < 0.10 when comparing WTC vs. ISO rats, and these DEGs also have an FDR < 0.05 when comparing WTC vs. Naïve rats—with each logFC having the same signs in both comparisons. No DEGs were identified under these two scenarios at days 1, 30, or 240, indicating that the WTC dust did not have a significant short-term effect.

At day 360, 14 DEGs satisfied these two scenarios (summarized in Table 1). These 14 DEGs were not significantly different between the ISO and Naïve rats; consequently, they could potentially reflect a long-term WTC dust effect beyond any attributable to ISO. Longitudinal analyses were also performed for the 14 DEGs by plotting the trajectories of logFCs (between WTC and ISO and between WTC and Naïve rats) across the time course. The mean logFCs were close to 0 at days 1 and 30, increased since day 240 and reached the peak at day 360. The result suggested that the attributable effect of WTC dust on the 14 DEGs were late-onset and persistent in the long term.

Several reports have shown that WTC dust exposures were significantly associated with increases in long-term cardiovascular disease risk among FR [3,4,42]. However, whether WTC dust inhalation is a causative agent for the cardiovascular disease or if the dusts may have affected (acutely or long term) cardiac gene expression remains unclear. In this study, among the 14 DEGs that were identified as being attributable to effects of the WTC dust, 12 (*Surf1*, *Tgfbr2*, *Ski*, *Pola2*, *P4hb*, *Rps18*, *Mfsd1*, *Scarb2*, *Fzd7*, *Safb*, *Glis1*, *Hspa8_2*) have been discussed in the literature as being associated with heart function. In particular, *Surf1* may be associated with mitochondrial biogenesis and COX (cytochrome o oxidase) activity in the heart of mice [43]. A common *TGFBR2* polymorphism in a human population was demonstrated to be associated with a risk of sudden cardiac arrest (SCA) due to ventricular arrhythmias (VA) in the setting of coronary artery disease (CAD) [44]. Additionally, in endothelial cells, TGFβ signals through the *Tgfbr2* receptor are essential for cardiac development and for maintaining cerebral vascular integrity [45]. Variants of *SKI* and *POLA2* are also associated with CAD risk at genome-wide level of significance [46]. Additionally, *Ski* localizes in the cell nucleus and can also deactivate cardiac myofibroblasts, probably by regulating the expression of *MMP* [47]. The transfer of *P4hb* to the heart of mice was reported to significantly reduce infarct size and cardiomyocyte apoptosis during cardiac infarction [48]. The expression of *Scarb2*, also known as *Limp-2*, has been found to be related to cardiac hypertrophy and heart failure in both rat and human myocardium. *LIMP-2* plays an important role in mounting the adaptive hypertrophic response to cardiac loading [49]. *Safb* was found to be crucial in cardiac fibrosis as it mediated the transcription of fibrosis-related genes. Further, *SAFB* has been considered to be a potential therapeutic target for cardiac fibrosis [50]. *Fzd7* has an important role in heart development. It controls angiogenesis and influences the retinal vascular development through the Wnt canonical pathway [51,52]. *Glis1* is also associated with cardiac development by regulating heart valve morphogenesis, and heart valves are crucial for ensuring that during contraction, blood flows in one direction from the heart to the lungs or to the rest of body [53]. Furthermore, *Glis1* potentially gets involved in the mechanism of mitral valve degeneration in mice and zebrafish [54]. Genetic variants of *HSPA8* gene changed its expression level, which could in turn affect the coronary heart disease (CHD) susceptibility in human [55]. The expression level of *Mfsd1* could be affected by left heart disease in rats [56]. *Rps18* may be associated with heart failure [57]. It is worth noting that among the 12 DEGs that were discussed above, the logFCs (WTC vs. ISO) of 9 DEGs have the same sign at both days 240 and 360 indicating potential long-term effects.

Since the WTC rats were actually exposed to a mixture of WTC dust and ISO, it is possible that the interaction between WTC dust and ISO could also influence the observed gene expression, particularly in the long term. In particular, ISO alone might provide some long-term protective effects, but such effects may not be detectable in the WTC group (i.e., the WTC dust plus ISO group) of rats due to potential ISO–WTC dust interactions. In addition to those 14 genes that have significant differential expressions between WTC and ISO and WTC and Naïve rats at day 360, this study further investigated potential long-term effects of WTC dust from the set of genes that have small p-values for WTC vs. ISO, but not necessarily WTC vs. Naïve which might reflect ISO–WTC dust interactions. Thus, by comparing WTC to ISO, 124 DEGs whose logFCs had the same sign at both days 240 and 360 with *p*-value < 0.05 at day 360 and *p*-value < 0.1 at day 240 were selected, where 60 of them were up-regulated in WTC and 64 of them were down-regulated.

Functional gene set enrichment analyses showed that the up-regulated DEGs were significantly enriched in regards to PPAR signaling pathways, these were characterized by changes in expression of *Aqp7*, *Fabp3*, *Rxrg*, *Ehhadh*, *Plin5* genes in here. Myocardial energy metabolism is important for cardiac structure/function, and PPAR proteins transcriptionally regulate myocardial energy metabolism [58,59]. Among the PPAR-associated genes up-regulated by the dust exposures here, *AQP7* is important in the maintenance of cardiac energy balance [60], and its expression is often seen to be up-regulated in pathologies where heart energy balance/substrate levels are altered [61]. Similarly, Retinoid X receptors (*RXR*α/β/γ isoforms, members of the superfamily of intracellular hormone receptors) are varyingly important in the regulation of isoform switching among myosin heavy chain proteins during aging [62], and are also implicated in regulation processes involved in any transitions from cardiac hypertrophic states to actual heart failure [63]. Several of the other genes found to be up-regulated, including fatty-acid-binding protein 3 (*FABP3*), enoyl-CoA, hydratase/3-hydroxyacyl CoA dehydrogenase (*EHHADH*), and perilipin 5 (*Plin5*), have also each been reported to play important roles in both normal cardiac function and cardiac disorders. For example, *FABP3* and *EHHADH* were up-regulated in the left atria of mitral regurgitation patients compared to in patients with aortic valve disease and normal controls [64]. *Plin5* regulates the formation and stabilization of cardiac lipid droplets (LD) associated with lipo-toxicity and tissue dysfunction in cardiac muscle [65]. It is thus not surprising that mice with over-expressed Plin5 exhibited a 3.5-fold increase in heart triglyceride content (vs. normal controls); over time, this cardiac excess of LD resulted in mild heart dysfunction, altered expression of PPARα-targeted genes, decreased mitochondrial function, and left ventricular concentric hypertrophia [65].

Whether or not associations exist between the identified DEGs and a risk for cardiac disease, as well as whether any of the up-/down-regulated genes might impart additive or multiplicative effects on this risk, is an interesting question. The design of the current study only focused on comparisons of the transcriptive profiles in the hearts of rats that underwent different exposures. These comparisons were made via differential expression analysis and functional enrichment analysis. It was hoped that these approaches would allow for determinations to be made to see if any of the identified DEG coded for proteins known to be associated with normal heart functions/dysfunction (based on existing literature) and/or whether these DEG coded for proteins could then impact on the activities of select pathways known to be critical for the maintenance of normal heart functions. Future translational studies are warranted to further address the issue of additive or multiplicative effects from the WTC dust exposure on the cardiac disease risk factors.

In the end, dysregulated expression of each of these genes could signal either initiation and/or progression of cardiac pathologies induced by WTC dust exposures, or reflect tissue responses to damage in the heart that had been induced earlier (before day 360) in the post-exposure period, or a combination of both. As previously noted, precisely how WTC dust-induced regulation of these groups of genes might ultimately be implicated in heart failure/other cardiac disorders in the FR remains to be determined.

## 5. Conclusions

The results of this study showed that repeated exposures to Ground Zero WTC dust were able to impact changes in gene expression in the hearts of exposed rats. More importantly, any ‘pure’ effects from the dusts did not manifest until a significant time had elapsed from the exposures themselves. Even > 20 years after the 9/11 disaster, this has potentially important implications for those FR who were exposed repeatedly to these dusts at Ground Zero during the first week after the buildings collapsed. Of note, the WTC dust was found to affect gene expression that could influence a variety of heart/cardiovascular functions, including coronary artery disease, heart development, cardiac fibrosis, myocardial hypertrophy, myocardial infarct, heart failure, etc. The WTC dust effect in the presence of potential ISO–WTC dust interactions could potentially impact the myocardial energy metabolism via PPAR signaling and heart valve development.

The authors are cognizant of several important limitations to these studies. As noted in the Methods, the co-exposure of the rats to ISO and the dust resulted in confounding outcomes. A new exposure method is being developed that will not require ISO, therefore it will allow for equivalent WTC dust delivery into the lungs, and will reiterate many of the outcomes seen here. The co-exposure effect has meant that the ISO might have negatively impacted (or conversely, amplified) the magnitude of any given WTC dust effect so that some dust effects could not be clearly discerned or defined as significant. It is also possible that WTC dust modulated any true effects of ISO on gene expressions. All attempts were made here to disentangle the confounding from the true effects of both the WTC dust and the ISO. Another limitation was that only a single dust dose that would approximate Ground Zero air levels of ≈250 mg dust/m^3^ was used. While dust levels were ≥1000 mg/m^3^ at Ground Zero in the first hours after the disaster, and a significant number of FR were exposed to such levels in that period, most of the exposed FR arrived on scene in the hours after when levels dropped to 100s of mg/m^3^ and stayed on-site for periods where air dust levels remained in this range. Further, exposures of rats to WTC dusts at levels that would reflect any FR exposures at ≥1000 mg/m^3^ have proven to be moreover lethal in the 2 h/day, 2 day paradigm that mimics the ‘reference FR.’ As a result, no useful long-term, post-exposure data from those rats could be obtained. Lastly, another key limitation here was that studies did not go further to determine if any observed effects manifest even beyond the 1 year post-exposure timepoint or the earlier effects persisted long term. The nature of this study was that several early timepoints (i.e., days 1 and 30) had to be evaluated to establish baseline outcomes (early-onset, short-lived, etc.). Based on the pool of results presented here, the next series of studies will have rats dedicated for the examination at far later timepoints (i.e., up to 2 year post-exposure) and only a few examined at those early timepoints (to confirm reproducibility of outcomes).

## Figures and Tables

**Figure 1 ijerph-19-00919-f001:**
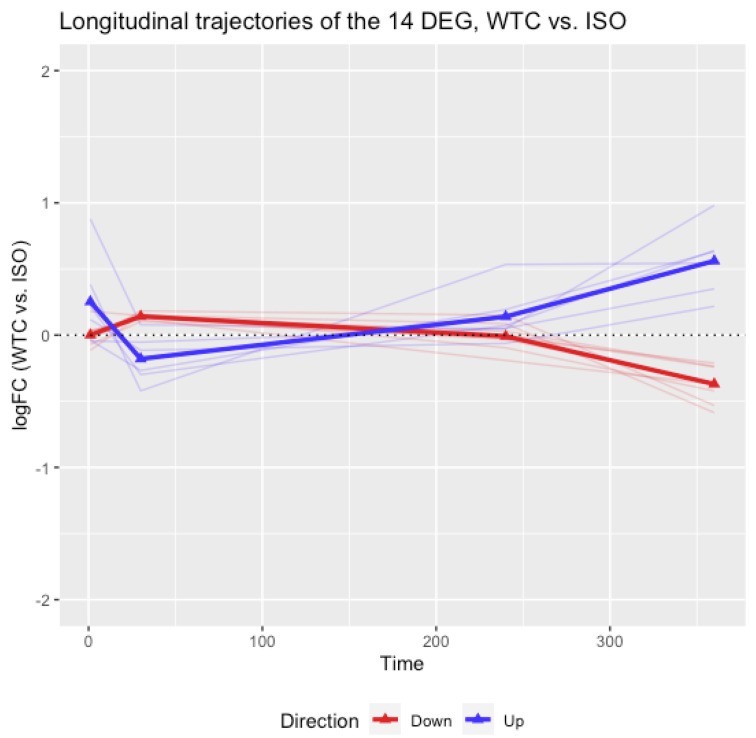
**The longitudinal trajectories of logFC between WTC and ISO for the 14 DEGs.** The 14 DEGs were identified at day 360 as listed in Table 1. The blue lines referred to DEG up-regulated at day 360, also labeled as ‘Up’, and the red lines referred to DEGs down-regulated at day 360, labeled as ‘Down’. The bold lines connecting the triangles are the mean trajectories.

**Figure 2 ijerph-19-00919-f002:**
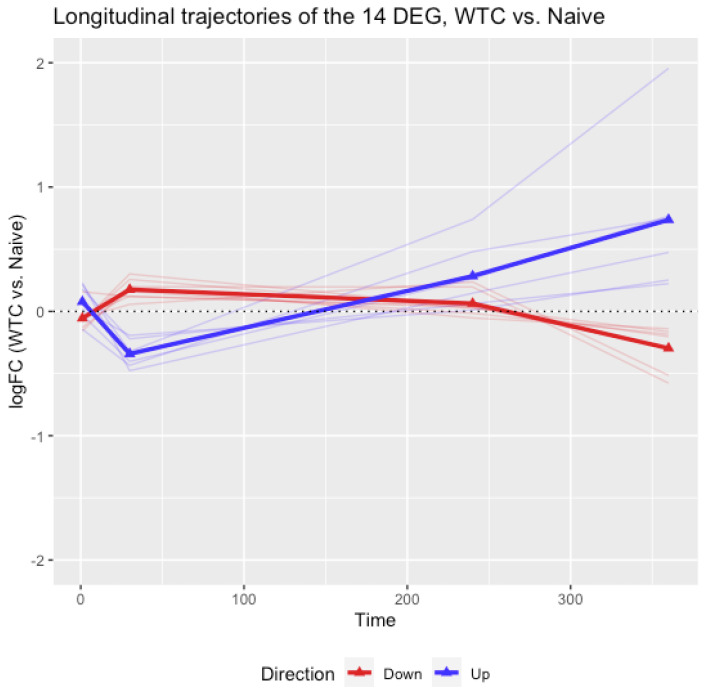
**The longitudinal trajectories of logFC between WTC and Naive for the 14 DEGs.** The 14 DEGs were identified at day 360 as listed in Table 1. The blue lines referred to DEGs up-regulated at day 360, also labeled as ‘Up’, and the red lines referred to DEGs down-regulated at day 360, labeled as ‘Down’. The bold lines connecting the triangles are the mean trajectories.

**Figure 3 ijerph-19-00919-f003:**
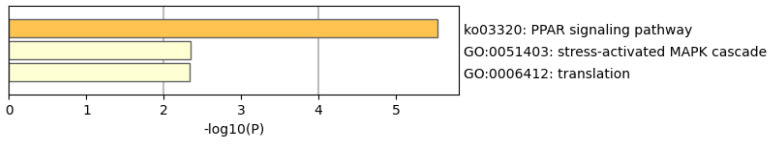
**Enrichment results of the selected 60 up-regulated DEG comparing WTC vs. ISO.** Functional gene set enrichment analysis was performed on the 60 DEGs that were up-regulated in WTC compared with ISO group at both day 360 (with *p* < 0.05) and day 240 (with *p* < 0.1).

**Figure 4 ijerph-19-00919-f004:**
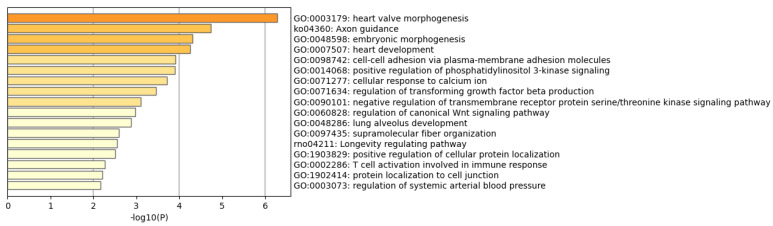
**Enrichment results of the selected 64 down-regulated DEG comparing WTC vs. ISO.** Functional gene set enrichment analysis was performed on the 64 DEGs that were down-regulated in WTC compared with ISO group at both day 360 (with *p* < 0.05) and day 240 (with *p* < 0.1).

**Table 1 ijerph-19-00919-t001:** The list of 14 DEGs reflecting attributable WTC effect on day 360.

	WTC vs. ISO			WTC vs. Naïve		
Name	logFC	*p*	FDR	logFC	*p*	FDR
Surf1	0.63	9.45 × 10^−6^	**1.53 × 10^−2^**	0.47	2.64 × 10^−4^	**3.48 × 10^−2^**
Tgfbr2	−0.42	1.93 × 10^−5^	**1.53 × 10^−2^**	−0.20	1.55 × 10^−2^	1.50 × 10^−1^
Ski	−0.37	3.01 × 10^−5^	**1.59 × 10^−2^**	−0.18	1.57 × 10^−2^	1.50 × 10^−1^
Pola2	0.35	5.54 × 10^−5^	**1.99 × 10^−2^**	0.22	4.08 × 10^−3^	9.65 × 10^−2^
P4hb	−0.24	2.12 × 10^−4^	**3.58 × 10^−2^**	−0.16	5.29 × 10^−3^	1.07 × 10^−1^
Rps18	0.54	4.28 × 10^−4^	**3.96 × 10^−2^**	0.75	1.36 × 10^−5^	**1.08 × 10^−2^**
Mfsd1	−0.21	4.77 × 10^−4^	**4.17 × 10^−2^**	−0.14	1.29 × 10^−2^	1.41 × 10^−1^
Scarb2	−0.36	5.17 × 10^−4^	**4.18 × 10^−2^**	−0.30	3.05 × 10^−3^	9.12 × 10^−2^
Fzd7	−0.59	7.53 × 10^−4^	**4.92 × 10^−2^**	−0.52	2.49 × 10^−3^	8.65 × 10^−2^
Isg20l2	−0.53	8.52 × 10^−4^	5.03 × 10^−2^	−0.58	3.76 × 10^−4^	**4.29 × 10^−2^**
Safb	0.22	9.69 × 10^−4^	5.16 × 10^−2^	0.25	2.34 × 10^−4^	**3.47 × 10^−2^**
Glis1	0.64	1.98 × 10^−3^	7.06 × 10^−2^	0.77	4.35 × 10^−4^	**4.53 × 10^−2^**
Hspa8_2	0.98	2.02 × 10^−3^	7.06 × 10^−2^	1.96	2.61 × 10^−6^	**7.07 × 10^−3^**
LOC294154	−0.24	2.07 × 10^−3^	7.19 × 10^−2^	−0.30	3.24 × 10^−4^	**3.83 × 10^−2^**

FDR is in bold if its value is less than the commonly used cut-off 0.05.

## Data Availability

Not applicable.

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
