# Peer review of "Longitudinal Impact of WTC Dust Inhalation on Rat Cardiac Tissue Transcriptomic Profiles"

_ijerph, 2022, doi:10.3390/ijerph19020919_

Round 1

Reviewer 1 Report

The content of the submitted manuscript is good but the presentation way of current form is not fulfilling the journal requirements. Modification is needed to consider for publication.

  - Title of the paper

The title of the paper looks good but in the same time, it can be modified to represent the manuscript in a better way.

Abstract

-          The abstract is not well written

-          You should include some of the main finding in the abstract section.

Abstract should have a conclusion of the study.

Introduction  

  • The objective of the study is also not clearly mention.
  • Add more on the basic of the problem in the introduction
  • More details about identified problems are required in the introduction section.
  • The author should focus mainly on the importance and significance of the study.
  • I suggest the author to demonstrate what does the paper add to the current literature? and what new knowledge is added by this study?

Add the unique of this study compared to other studies discuss the same issue.

-Discus merits and limitations of technique applied.

 Material and Methods

-     The material and method section is too weak in the manuscript and you need to focus on it more.

Result and discussion

  • The presentation fails to discuss the summary, and trying to some of vague reason which is not the explanation.
  • The explanation for the critical analysis is not sufficient, although some of the good points have has been identified.

Conclusion

-          Please rewrite the conclusion with the proper explanation in the R & D.

Other comments:

English editing is needed in some parts of the manuscript.
Abbreviations should be explained before the introduction.

Reviewer 2 Report

1.Interesting topic and study, impressive scientific work . 2. Suggestion to add some long term epidemiological data referring to cardiovascular diseases among FR from WTC. 3. Could you make any comparison with other known exposure of rescue personnel and health impact of respiratory toxicants, despite not having similar disaster to compare ? 4. Study limitations and strengths have to be added. 5. Perspectives and relevance for medical practice may be outlined

Reviewer 3 Report

Dear editor and authors

The study presents interesting results of experiments, which can shed light on the causes of some illnesses in professionals exposed after the event that occurred on September 11, 2001. The study was well conducted and the manuscript is well written. I suggest minor review before publication in IJERPH.

Line 16: authors must include the meaning of WTC the first time it is mentioned.

Lines 70-72: it is not appropriate to present results in the introduction. I suggest mentioning that this is the first study to evaluate such an outcome... that the study has the potential to demonstrate such evidence etc.

Lines 93-94: authors could include a table summarizing the physicochemical characteristics of the WTC53. For example, presence of potentially toxic elements, organic compounds and percentage of PM10, PM2.5 and PM0.1. The presentation in table format can make the data clearer compared to the current presentation in lines 101-103.

Conclusions: authors should emphasize study limitations in the conclusions

Reviewer 4 Report

Overall the study provides important information regarding the potential effects of WTC dust on the CVD system. The authors found that the long-term trajectories of animals models exposed to WTC dust were significantly different than those treated with ISO alone. The high scientific merit of this study is important to future studies of environmental pollutants and heart disease. However, the paper requires edits to restructure the order of the information presented, as well as clarification about how the outcome logFC was operationalized.

While the authors mention that the genes and pathways of interest are important to heart function, they do not list the exact conditions that would result from changes to these genes that have also been noted in the human epidemiologic literature. I would like to see more linkage between the epidemiologic studies and the laboratory study acknowledged in the introduction to this paper.

Cardiac diseases in adults can result from a multitude of processes including arterial stiffening, ion channel disruption, and an overall weakening of the heart tissue leading to cardiomyopathy or heart failure. I would like to see more discussion about what heart diseases may be caused by alterations in these genes that are relevant to the WTC workers. This information is provided in the discussion (Lines 410 - 442) and I would like to see some of this moved into the introduction and results.

I would like more discussion about the conceptualization of the outcome of logFC of the 14 genes studied. Is the logFC a global measure of the changes in the 14 genes? Or was the logFC taken for each individual gene? If this is the case, how was Figure 2 created?

The authors mentioned using PCA to look for patterns in the changes in gene expression. I would like to see the results of the PCA as well as a description of the patterns that the PCA found within those 14 genes. For example, were certain clusters of genes up-regulated while others were down-regulated? Could concurrent up or down regulation of multiple genes have additive or multiplicative biological impacts on risk for cardiac disease?

Lines 218 to 258 appear to belong in the methods section rather than the results, and certain sections of lines 300 to 336 also appear to describing methods rather than results.

Table 1 needs subscripts to define how the reader should interpret the bolded FDR values. It would be helpful to understand if the differences were due to up-regulation or down-regulation of the genes in the table.

Does Figure 2 describe the trajectory of a certain number of genes studied? Or is this a summary variable for the 14 genes in the literature? Please show the standard errors around the mean values for the trajectories. The figure looks like the line was drawn between the mean for each time point. Is it possible to fit a spline or Joinpoint model to this data? that way you can identify significant inflection points over time rather than using a qualitative visual assessment of the changes over time.

In line 330, the authors suggest that the PPAR pathway appears to be of importance. This information nor the important of the PPAR pathway was not provided in the introduction. If the authors were formally testing for activation of the PPAR pathway, please provide that information in the methods. Otherwise this belongs in the discussion section.
